# Advanced Analytical Methods of the Analysis of Friction Stir Welding Process (FSW) of Aluminum Sheets Used in the Automotive Industry

**DOI:** 10.3390/ma16145116

**Published:** 2023-07-20

**Authors:** Krzysztof Chyła, Krzysztof Gaska, Anna Gronba-Chyła, Agnieszka Generowicz, Katarzyna Grąz, Józef Ciuła

**Affiliations:** 1Faculty of Energy and Environmental Engineering, Silesian University of Technology, 44-100 Gliwice, Poland; krzysztof.chyla@polsl.pl (K.C.); krzysztof.gaska@polsl.pl (K.G.); 2Faculty of Natural and Technical Sciences, John Paul II Catholic University of Lublin, ul. Konstantynów 1 H, 20-708 Lublin, Poland; amgronba@kul.pl (A.G.-C.); katarzyna.graz@kul.pl (K.G.); 3Department of Environmental Technologies, Cracow University of Technology, ul. Warszawska 24, 31-155 Cracow, Poland; 4Faculty of Engineering Sciences, State University of Applied Sciences in Nowy Sącz, Zamenhofa 1A, 33-300 Nowy Sącz, Poland; jciula@pwsz-ns.edu.pl

**Keywords:** friction stir welding, mechanical properties, energy reduction, strength tests, friction welding process

## Abstract

The paper provides general information on selected methods of joining aluminum sheets. The main focus is on the strength of the friction stir welding connection and the energy consumption of the process. The practical part of the study used aluminum alloy 2024-T3, the most commonly used alloy in the automotive industry. The study consisted of the FSW welding of two pieces of overlapping sheet metal, using different process parameters. The thickness of the sheet used was 1 mm. After the welding was completed, the test specimens were broken on a testing machine. During the tests, the appropriate process parameters were selected at which the weld showed the highest strength. The effect of implementing the FSW process should be to increase the efficiency of sheet-metal joining. It should also result in a reduction in the energy intensity of the process, which will translate into the lower production cost of the final product. Strength tests were carried out on eighteen samples of joined sheets. The best results were obtained at a feed rate of 100 (mm/min) and a rotational speed of 900 (rpm). It can also be seen that friction welding is an efficient and low-emission way of joining metals. Through the analysis, it can be concluded that in order to perform one meter of satisfactory welding, CO_2_ emissions will be approximately 310 g. These are calculations based on data published by the National Balancing and Emissions Management Centre from 2019. Analyzing the 2019 data from the Society of Automobile Manufacturers, it is safe to say that the potential for implementing the FSW method in the automotive industry is huge.

## 1. Introduction

Friction stir welding is one of the most modern methods of joining metals and their alloys in the solid state [1]. However, ensuring the constancy of the performance of the joints requires the optimal choice of welding-process parameters, i.e., tool speed and feed rate [2]. This method was developed and patented in 1991 by the English Welding Institute [3]. The method makes it possible to make high-quality joints of metallic materials [4]. Observations of microstructures show fine grains and a limited Heat-Affected Zone limiting crack growth [3]. The obvious advantages of using this technology to join sheets of different metallic materials make it widely used by the automotive and shipbuilding industries [5]. The repeatability and reliability of the process, combined with its ability to assemble lightweight alloys, are attracting aerospace applications, and the use of aluminum alloys continues due to the continuous improvements made in their production by the industry [6,7]. As long as these alloys are used in aerospace applications, FSW will continue to attract industry interest [8,9]. The best joining results are obtained by conducting the welding process with controlled clamping force and with the correct setting of the inclination angle and depth of its immersion in the material [10].

The effectiveness of the resulting joint is influenced by the geometrical parameters; the height and shape of the pin and the tool’s thrust surface affect both metal flow and heat generation under frictional forces [11]. In addition, the force applied to the rotating tool during the process itself must be appropriately selected, as the pressure generated at the tool’s thrust surface and under the pin tip determines the heat generation during the process [10]. The rotational speed and feed rate must also be selected appropriately in order to obtain efficient joints [12]. In friction stir welding (FSW), a rotating tool with a given profile moves forward along the weld line [12,13]. Frictional contact between the FSW tool and the workpiece and plastic dissipation are responsible for heat generation and material softening [14,15]. As the tool moves forward, a weld is formed by material mixing [16]. The heat generated during friction occurs mainly under the shoulder due to the larger surface area. Contact conditions between the protrusion and the workpiece can be of the sliding or sticky type depending on the value of the tangential shear strength [17]. This strength is a function of temperature and strain rate. At first glance, the process is simple, as it involves only the movement of the tool through the weld line. However, the actual mechanism of FSW is complex, as it is highly non-linear and coupled [18,19]. The strong coupling between the temperature field and material behavior does not allow researchers to rely on simple tribological tests to represent the frictional behavior in FSW [20]. Either measurements must be made during the actual FSW execution (which is not trivial) or numerical simulations must be performed [18].

The mixing zone of the weld was made up of very small and evenly spaced grains with an average diameter of a few micrometers. A difference was observed between the hardness profiles for low and high tool speeds: the profiles for lower speeds were characterized by only one minimum, and for higher speeds, by two minima [21,22,23]. The use of heating leads to a levelling of the plastic deformation of FSW-welded thin-walled sheets [24,25]. This heating increases the temperature of the process, allowing higher feed rates to be used and, thus, reducing the process time [26]. Tests of a joint made with the same parameters have shown a significantly higher tensile strength for heated welding [27]. One of the most important advantages of friction stir welding (FSW) is the lower energy consumption compared to other material joining processes. Zifcak et al. obtained the highest tool life for a speed of 400 rpm and a welding speed of 55 mm/min. However, then a 100% correct joint was not achieved [22]. The static tensile test is the most common test used in materials strength and materials engineering. The primary purpose of the tensile test is to test the basic strength parameters of the materials being analyzed, i.e., the elastic limit, yield strength, or maximum strength of the material [28]. The test is relatively simple and is carried out on universal testing machines. Pre-prepared specimens of suitable shape and dimensions are tested [29]. Most commonly, both flat and round machined specimens are used; sometimes unmachined materials are also used. Flat samples come in two varieties, with or without heads. In this case, the most important size is their initial thickness [30,31]. The specimens thus prepared are placed in special grips located on the machine and, once accurately positioned, are subjected to an axial load, with the force acting smoothly from the initial stage of the test until the specimen breaks abruptly [30].

The course of the test is recorded on a computer, using mounted recorders and strain gauges. From the data read, a tensile graph is formulated for the specimen in question. The graph is highly dependent on the material used, with the most common division being brittle and ductile [32]. By carrying out a static tensile test, we are able to read off a few characteristic points on the diagram that are very important for determining the basic strength characteristics of the material being tested. The stresses occurring at these characteristic points have appropriate names and describe specific properties of the specimen [30,33].

The manufacturing sector is currently facing a global need to reduce the environmental impact of human activities [31]. As manufacturing is responsible for a significant proportion of global CO_2_ emissions, research should be directed towards understanding the environmental impact of production processes and, in doing so, realizing their full potential in reducing overall CO_2_ emissions [34,35]. From the breakdown analysis provided by the IEA, it can be seen that industry plays an important role and accounts for almost 40% of total consumption [36,37]. Specifically, in the industrial sector, CO_2_ emissions are due to both direct and indirect emissions. The latter are due to electricity use and currently account for 18% of the total [38,39]. Improving energy efficiency is an important strategy in relation to the security of energy supply, climate change, and competitiveness, and it can be achieved through technological changes or better organizational management or behavioral changes [40,41]. Technological advances save electricity and present carbon-reduction effects [42,43,44]. However, in practice, new and old technologies coexist in organizations, and investment in innovative technologies is usually discouraged due to high costs [45,46]. By using the innovative FSW method on a large scale in the automotive industry, the energy consumption needed to make sheet metal joints will be significantly reduced, and, consequently, CO_2_ emissions will be reduced [47].

The studies available in the literature on the FSW method address the influence of parameters and tool geometry on the mechanical properties and microstructure of joints welded by material displacement. However, there are little data on the environmental impact of the process, taking into account CO_2_ emissions in relation to the strength of the joint by comparing the tool speed and feed rate.

## 2. Materials and Methods

As part of the research work carried out, a scientific and technological problem was solved concerning the optimization of the technological process of welding aluminum sheets with the FSW method, using original analytical methods, including strength tests. The result of the performed research was the selection of optimal process parameters in terms of weld strength. Based on the analysis of the research results obtained, a tool for the friction stir welding of aluminum sheets was designed. Within the scope of the work undertaken, the following is proposed:-Selection of the materials, tools, and machinery needed to carry out the test:

The test is operated with a specially designed tool (Figure 1) with a rotating plain mandrel over the contact area of the pressed tiles. Similar tools are available on the market, but tools with such a thin working tip are not available. Thanks to the use of the ultra-heavy method, it is possible to perform less effective welding of the material.

The friction welding process was carried out on an FWF 32J2 JAFO JAROCIN universal milling machine produced by Marat Sp. z o.o. Jarocin, Poland (Figure 2), with a motor power of 2.2 kW and a feed motor power of 0.25 kW. The parameters studied during the experiment were the tool speed and feed rate.

-Selection of process parameters to obtain high tensile strength (feed and spindle speed; see Table 1).

The total thickness of the plates to be joined was 2 mm, while each individual plate had a thickness of 1 mm. A photograph of the tests performed is presented below (Figure 3). This photo shows the start of the welding process and the fixture used to fix the plates being joined.

-Conducting laboratory tests using the testing machine:

The strength tests included a static tensile test, and the tests were carried out on a Zwick/Roell Z250 kN ripper produced by Zwick/Roell, Haan, Germany. After the welding process, the specimens were aged naturally (seasoned) for 7 days before strength testing. The specimens were placed in the jaws of the machine, held by frictional forces between the jaws and the gripping parts of the specimen.

The strength test was performed according to the PN-EN ISO 6892-1 norm, with the following parameters:-Initial tension force 5 N,-Crosshead speed 12 mm/min.

The picture below (Figure 4) shows the basic parameters of the tensile tester.

-Analysis of the test results obtained and conducting a statistical study using the Hartley method. The three-level plan allows for a number of experiments equal to the combination of input factors at all levels of variation, which can be written as 3*n*. A disadvantage of this test plan is that the number of experiments required increases rapidly as the number of input factors increases. For this reason, the three-level plans used in practice usually do not exceed the number of input factors, *n* = 3.

General formulae for the individual regression coefficients:$$\begin{array}{c} b_{0}=\frac{1}{9}\left(-y_{1}+{2y}_{2}-y_{3}+{2y}_{4}+{5y}_{5}+{2y}_{6}-y_{7}+{2y}_{8}-y_{9}\right)\\ b_{k}=\frac{1}{6}\sum_{i=1}^{n}x_{1}{ȳ}_{i}\\ b_{11}=\frac{1}{6}\left(y_{1}+y_{2}+y_{3}-{2y}_{4}-{2y}_{5}-{2y}_{6}+y_{7}+y_{8}+y_{9}\right)\\ b_{22}=\frac{1}{6}\left(y_{1}-{2y}_{2}+y_{3}+y_{4}-{2y}_{5}+y_{6}+y_{7}-{2y}_{8}+y_{9}\right)\\ b_{12}=\frac{1}{4}\left(y_{1}-y_{3}-y_{7}+y_{9}\right) \end{array}$$

-Calculation of the average electricity consumption consumed in the material welding process;-Analysis of the impact of electricity consumption.

## 3. Results and Discussion

The object of the research was to select the optimum parameters for the FSW friction welding process in order to achieve the best strength properties of the joint, in particular, the tensile strength of the 2024 T3 aluminum alloy sheets. Another sub-goal was to test the environmental impact, the energy consumption used to make the joint. The sheets to be joined were 1 mm thick, and an overlap joint made by friction stir welding was analyzed. The test schedule for the FSW method was carried out using the Hartley plan (PS/DK 32) [47].

Alloy 2024 T3 has very good mechanical properties and wear resistance of the material; therefore, it is used in the automotive and aerospace industries. The mechanical properties are shown in Table 2 [33].

The symbol T3 indicates the basic state of the product. The alloy was subjected to heat treatment, including a supersaturation process, followed by cold deformation and, finally, natural ageing to obtain a stable product state. The chemical composition according to European Standard EN 573-1 is presented in the table below (Table 3) [33].

The samples for strength tests were cut from FSW-jointed specimens in the direction perpendicular to the joint line. The width of the specimens was 12.5 mm. FSW joining tests included two specimens each joined together using the same parameters; the FSW joint-strength test results are shown in Table 4.

Based on the analysis of the test results obtained, it is concluded that the best values of tensile strength in the FSW method were achieved for the following values: rotational speed of 900 (RPM) and feed rate of 100 (mm/min). These are 4.1 kN and 4.09 kN, respectively. The best tensile strength results are summarized in the graph below (Figure 5).

## 4. Statistical Test for the Hartley Plan FSW Method

Table 5 presents the tensile test results for selected specimens made of aluminum alloy 2024 T3 joined by friction stir welding.

Table 6 presents the distribution of values for the ordered Hartley plan matrix. The values in the table are ordered from the highest rotational speed to the lowest. Each of them was assigned an appropriate feed value.

Table 7 contains information about the breaking strength, F_1_ and F_2_. On this basis, the average breaking strength for each case was calculated.

The input factors of the process under study are presented below:

X_1_—rotational speed V, presented in the range from 900 to 1700 rpm.

X_2_—the value of the feed rate *p*; the value varies from 60 to 100 mm/min.

In order to carry out the experiment, it will be necessary to perform some of the most important calculations, which are presented in turn below.

Calculation of central values, in other words, input factor values.
(1)$$X_{10\;}=\frac{Vmax+Vmin}{2}=\frac{1700+900}{2}=1300$$
(2)$$X_{20\;}=\frac{pmax+pmin}{2}=\frac{100+60}{2}=80$$
where Vmax—maximum rotational speed; Vmin—Minimum rotational speed; pmax—maximum feed speed; and pmin—minimum feed speed.

Calculation of the units of variation:(3)$$∆x_{1=}\frac{Vmax-Vmin}{2}=400$$
(4)$$∆x_{2=}\frac{Pmax-Pmin}{2}=20$$

Coding of factors:(5)$$x_{1=}\frac{x_{1}-x_{10}}{∆x_{1}}=\frac{V-1300}{400}$$
(6)$$x_{2=}\frac{x_{2}-x_{20}}{∆x_{2}}=\frac{p-80}{20}$$
y = Ft(7)

Based on the interaction parameters obtained, a planning matrix is created (Table 8). The + and − symbols found in column x_1_ x_2_ are obtained by multiplying columns x_1_ and x_2_. On the basis of the table shown, following the same procedure, a matrix consisting of more input factors can be created.

The table presented (Table 8) above shows the planning matrix with the contrast that determines −1 = x_1_x_2_ and the results from the rupture force measurements and the average of the two measurements.

Using the formula:(8)$$S^{2}{\left(y\right)}_{i}=\frac{\sum_{i=1}^{r}{(y}_{ui}-{ȳ}_{i}{)}^{2}}{r-1}$$
$$S^{2}{\left(y\right)}_{1}=\frac{(3.62-3.69{)}^{2}+(3.74-3.4578{)}^{2}}{2-1}=0.0036$$

The values of the measurement error variance in the subsequent experiments were calculated, and the results are summarized in the table (Table 9).

Checking the repeatability of the experimental conditions:-Calculating the G-factor on the basis of the experiments carried out:
(9)$$G=\frac{S^{2}\left(Y\right)imax}{\sum_{i=1}^{N}S^{2}(y{)}_{i}}=\frac{0.015625}{0.044325}=0.353\;S^{2}\left(Y\right)imax-highestvaluefromtable\;8$$

-Calculation of degrees of freedom:

f_1_ = *n* = 9(10)

f_2_ = r − 1 = 2 − 1 = 1(11)

-The value of the Gkr coefficient was selected from the tables based on the calculated degrees of freedom:

G_kr_= G_(a:f1:f2)_ = G_(0.05:9:1)_ = 0.6385(12)


(13)
$$G<G_{kr}—\mathrm{This}\;\mathrm{means}\;\mathrm{that}\;\mathrm{the}\;\mathrm{conditions}\;\mathrm{for}\;\mathrm{the}\;\mathrm{performance}\;\mathrm{of}\;\mathrm{the}\;\mathrm{experience}\;\mathrm{can}\;\mathrm{be}\;\mathrm{considered}\;\mathrm{to}\;\mathrm{be}\;\mathrm{reproducible}.$$


## 5. Calculation of Regression Coefficients Based on Table 8 and Table 9

Calculated individual values of regression coefficients based on general formulae for regression coefficients:(14)$$\begin{array}{c} b_{0}=\frac{1}{9}\left(-y_{1}+{2y}_{2}-y_{3}+{2y}_{4}+{5y}_{5}+{2y}_{6}-y_{7}+{2y}_{8}-y_{9}\right)\\ \;\;\;\;\;=\frac{1}{9}\times (-3.69+2\times 3.74-3.45+2\times 3.16+5\times 3.295+2\times 3.5\\ \;\;\;\;\;-4.1+2\times 3.85-3.68)=3.34 \end{array}$$
(15)$$b_{1}=\frac{1}{6}\sum_{i=1}^{9}x_{1}{ȳ}_{i}=\frac{1}{6}\times -0.75=-0.1$$
(16)$$b_{2}=\frac{1}{6}\sum_{i=1}^{9}x_{2}{ȳ}_{i}=\frac{1}{6}\times 0.34=0.057$$
(17)$$\begin{array}{c} b_{11}=\frac{1}{6}\left(y_{1}+y_{2}+y_{3}-{2y}_{4}-{2y}_{5}-{2y}_{6}+y_{7}+y_{8}+y_{9}\right)\\ \;\;\;\;\;=\frac{1}{6}\times (3.69+3.74+3.45-2\times 3.16-2\times 3.295-2\times 3.5\\ \;\;\;\;\;+4.1+3.85+3.68)=0.43 \end{array}$$
(18)$$\begin{array}{c} b_{22}=\frac{1}{6}\left(y_{1}-{2y}_{2}+y_{3}+y_{4}-{2y}_{5}+y_{6}+y_{7}-{2y}_{8}+y_{9}\right)\\ \;\;\;\;\;=\frac{1}{6}\times (3.69+2\times 3.74+3.45+3.16-2\times 3.295+3.5+4.1\\ \;\;\;\;\;-2\times 3.85+3.68)=2.46 \end{array}$$
(19)$$b_{12}=\frac{1}{4}\left(y_{1}-y_{3}-y_{7}+y_{9}\right)=\frac{1}{4}\times \left(3.69-3.45-4.1+3.68\right)=-0.045$$

The significance of the regression coefficients was then assessed.

-Calculation of measurement error variances:


(20)
$$S^{2}\left(y\right)=\frac{1}{N}\sum_{i=1}^{N}s^{2}(y{)}_{i}=\frac{1}{9}\times 0.044325=0.004925$$


-Calculation of the number of degrees of freedom:

f = N(r − 1)=9 × (2 − 1) = 9(21)

-Determination of the critical value of the t_kr_ coefficient on the basis of tabulated data:

t_kr_ = t_(a:f)_ = t_(0.05:9)_ = 2.2619(22)

-Determination of the coefficient value b_0kr_:


(23)
$$bkr=tkr\sqrt{\frac{a}{Nr}}s^{2}\left(y\right)=2.2619\times \sqrt{\frac{4.2002}{9\times 2}}\times 0.004925=0.0054$$


Here, ${{|b}_{0}|>b}_{kr}$, so the coefficient $b_{0}$ is taken into account;$\;{{|b}_{1}|>b}_{kr}$, so the coefficient $b_{1}$ is taken into account;$\;{{|b}_{2}|>b}_{kr}$, so the coefficient $b_{2}$ is taken into account;$\;{{|b}_{11}|>b}_{kr}$, so the coefficient $b_{11}$ is taken into account;$\;{{|b}_{22}|>b}_{kr}$, so the coefficient $b_{22}$ is taken into account; and ${{|b}_{12}|>b}_{kr}$, so the coefficient $b_{12}$ is taken into account.

After eliminating expressions that have no impact, the regression equation is as follows:(24)$$ŷ=3.34-0.125x_{1}+0.057x_{2}-0.43x_{1}^{2}-2.46x_{2}^{2}-0.045x_{1}x_{2}$$

Checking the validity of the regression equation:-Calculation of the adequacy variance:
(25)$$s_{ad}^{2}=\frac{r\sum_{i=1}^{N}(ȳi-Ўi{)}^{2}}{N-k-1}=\frac{2\sum_{i=1}^{9}(ȳi-Ўi{)}^{2}}{9-2-1}=\frac{2x0.004925}{6}=0.001642$$

-Determining the number of degrees of freedom of the numerator:

f_l_ = f_1_ = N − k − 1 = 6(26)

-Determining the number of degrees of freedom of the denominator:

f_m_ = f_2_ = N × (r − 1) = 9 × 1 = 9(27)

-Definition of value F_kr_:


(28)
$$F_{kr}=G_{\left(0.05;6;9\right)}=3.3738$$


-Determination of the empirical value, F:


(29)
$$F=\frac{s_{ad}^{2}(y)}{s^{2}(y)}=\frac{0.001642}{0.004925}=0.3334$$


$F{<F}_{kr}$; hence, the regression equation obtained is adequate, and its significance level is α=0.05.

Decoding the regression Equation (17):(30)$$F_{u}=3.34+0.125(\frac{V-1300}{400})+0.057(\frac{p-80}{20})-0.43{\left(\frac{V-1300}{400}\right)}^{2}-2.46{\left(\frac{p-60}{20}\right)}^{2}-0.045\left(\frac{V-1300}{400}\right)\left(\frac{p-80}{20}\right)$$

In Mathematica, the decoded regression equation was simplified to the following form:$$-41.7811+0.994163\times p-{0.00615\times p}^{2}+0.00775\times v-5.625\times 10^{-6}\times p\times v-2.6875\times 10^{-6}\times v^{2}$$

Subsequently, the variables for feed rate and spindle speed were determined: for feed rate, *p* (60, 80, 100) (mm/min); and for rotational speed, v (900, 1300, 1700) (RPM).

Decoding the regression equation requires substituting the previously coded input factors into the resulting equation. The final equation containing the values of the parameters changed during the experiment illustrates the effect of spindle speed and milling machine table feed rate on the strength of the weld formed.

The obtained regression equation can also be used for the graphical interpretation of the results by creating a three-dimensional graph showing the influence of the studied process parameters on the maximum value of the force transferred through the joint (Figure 6).

In the next step, the average power consumption needed to make one millimeter of weld was calculated. The joint with the best tensile strength was selected for the calculation (Table 10).

The energy consumed to weld 1 mm of material was calculated from the following formula:E = W × t [kWh](31)
where W is the rated power of the machine, and W = motor power + feed motor power; and t is the time.

The time required to join 1 mm of the test sheet was calculated from the following formula:(32)$$t=\frac{l}{Vf}\times 60=\frac{1}{100}\times 60=0.6\;s=0.000167\;h$$
where l is the weld length, and Vf is the tool feed rate.

Energy for welding 1 mm of weld:E = 2.45 × 0.000167 = 0.00040915(33)

For example, to obtain 1 m of a satisfactory weld based on the above calculations, it is necessary to use 0.4 kWh.

## 6. Conclusions

The research work carried out solved a scientific and technological problem concerning the optimization of the technological process of FSW welding of aluminum sheets, using proprietary analytical methods, including strength tests. A comparative graph (Graph 1) taking into account a change in the value of individual welding parameters in relation to the maximum force transferred through the joint illustrates that a change in the value of the parameters in the examined variation range has a non-linear effect on the strength of the joint. Data collected from strength tests of friction-welded joints and calculated data calculated from the obtained regression equation present very similar values. On the basis of Graph 2, the highest probability of values is observed for the feed value of 80 mm/min, while for the values of 60 and 100 mm/min, a large statistical error can be observed. The high similarity of the results allows us to conclude that the mathematical model of the process takes into account the influence of the individual friction welding parameters on the quality of the connection to a fairly good extent. Based on the strength tests carried out for 2024 T3 aluminum alloy sheets, it can be observed that joints realized with the parameters of a feed rate of 100 mm/min and rotational speed of 900 rpm have the best joint strength properties, i.e., of 4.1 kN. The joints are characterized by good quality and decent mechanical properties. The tests of lap joints obtained with different tool speeds (900, 1300, and 1700 rpm) and feed rates (60, 80, and 100 mm/min) showed that adequate joint quality in terms of strength can be achieved with a wide range of process parameters. All speeds and feed rates used produced homogeneous joints with no visible defects. As can be seen, by modifying the welding parameters appropriately, other joint-strength values can be obtained. The adopted process parameters, i.e., the rotational speed of the tool and the feed rate, have the greatest impact on the structure and properties of the joint welded using the FSW method. In the conducted tests, the parameters that were changed were the feed speed of the milling table and the rotational speed of the spindle; the pressure force of the tool and its depth in the material were unchanged during all welding tests. All combinations of welding process parameters made it possible to create a weld with satisfactory strength parameters. It follows that the correct structure of the joint and the appropriate value of the weld strength can be obtained in a wide range of changed process parameters. In the conducted tests, the feed speed was changed in the range from 60 to 100 mm/min, and the rotational speed of the tool was in the range from 900 to 1700 rpm; the angle of inclination of the tool head was constant for all welding tests and amounted to 3°. The tool with a simple shank design used in the tests was sufficient to create a joint with good mechanical properties in the tested range of changed parameters for the tested aluminum alloy. In the future, tests of joints made using the butt method are planned to verify the repeatability of the process and the correctness of the calculation of the mathematical model. In the next stages of the research, the environmental impact of the process will be taken into account, with a particular emphasis on reducing CO_2_ emissions.

## Figures and Tables

**Figure 1 materials-16-05116-f001:**
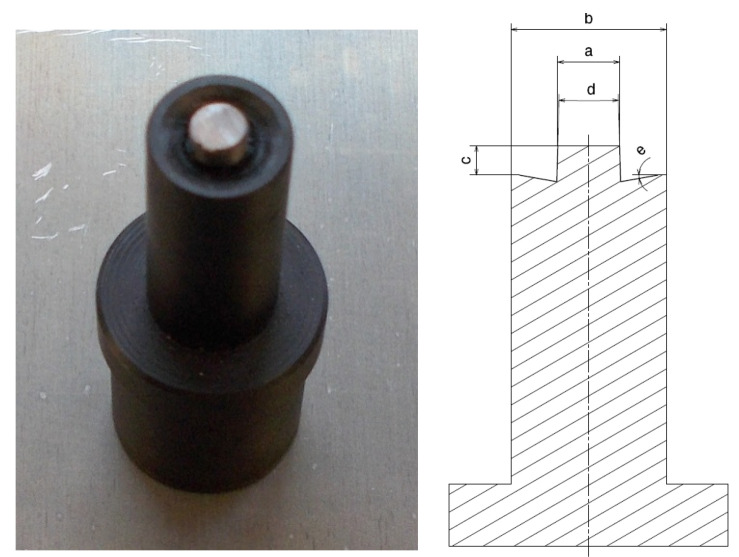
Photo showing the tool used for friction welding. Parameters of the tool presented: (a) tool pin cylindrical, diameter M4; (b) tool collar cylindrical, diameter of 10 mm; (c) pin height, 1.9 mm; (d) head angle, 3%; (e) flange contact surface to be welded is conical with a concave angle of 10 degrees.

**Figure 2 materials-16-05116-f002:**
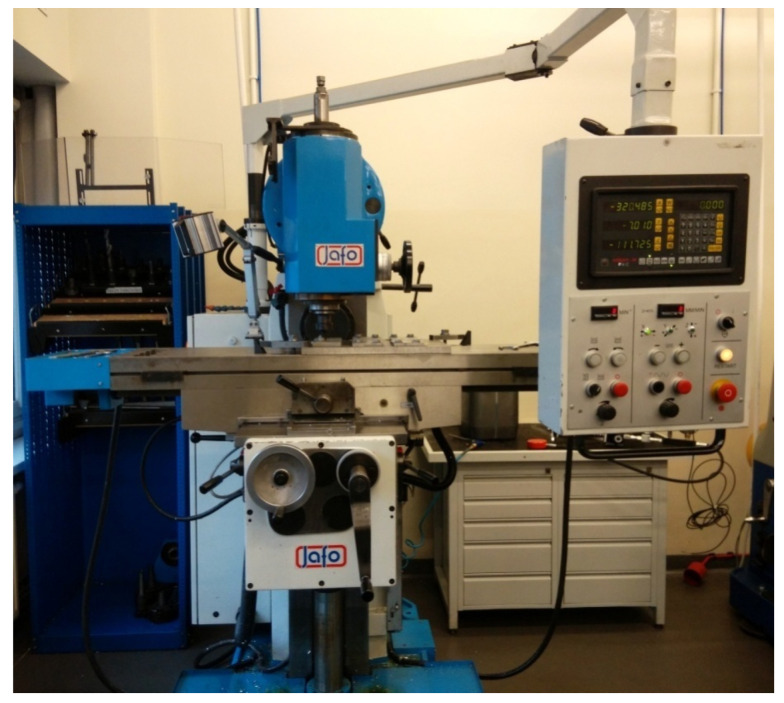
Universal milling machine FWF 32J2 JAFO JAROCIN.

**Figure 3 materials-16-05116-f003:**
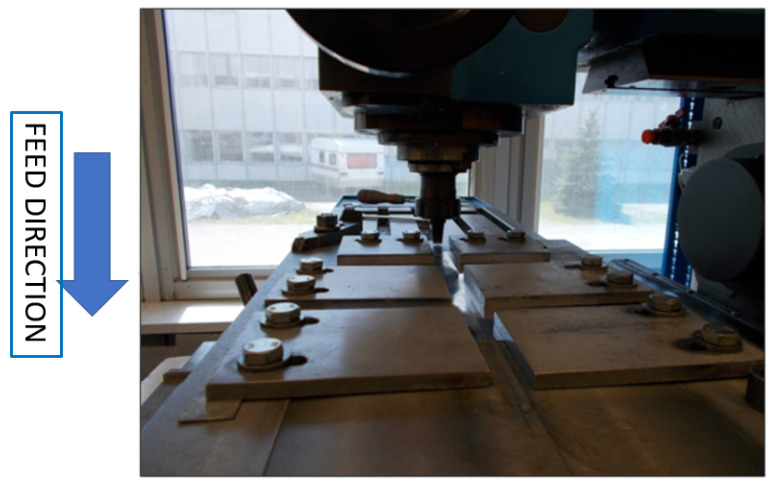
Start of the friction welding process.

**Figure 4 materials-16-05116-f004:**
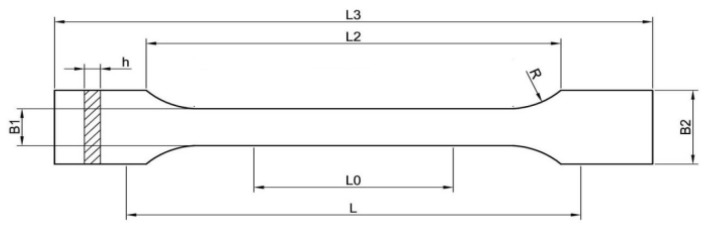
Tensile tester. L3—total length, 150 mm; R—radius, 60 mm; L2—the distance between the wide parallel parts, 106 mm; B2—width at the ends, 20 mm; B1—width of the narrow part, 10 mm; H—recommended thickness, 2 mm; L0—measuring length, 50 mm; L—initial distance between handles, 115 mm.

**Figure 5 materials-16-05116-f005:**
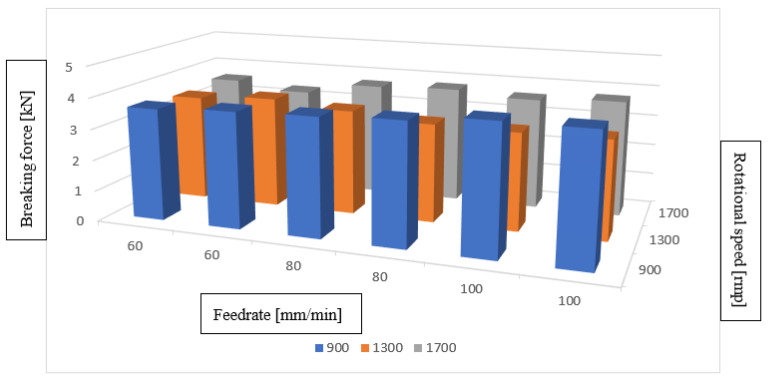
Summary of the best tensile strength values obtained using the FSW method.

**Figure 6 materials-16-05116-f006:**
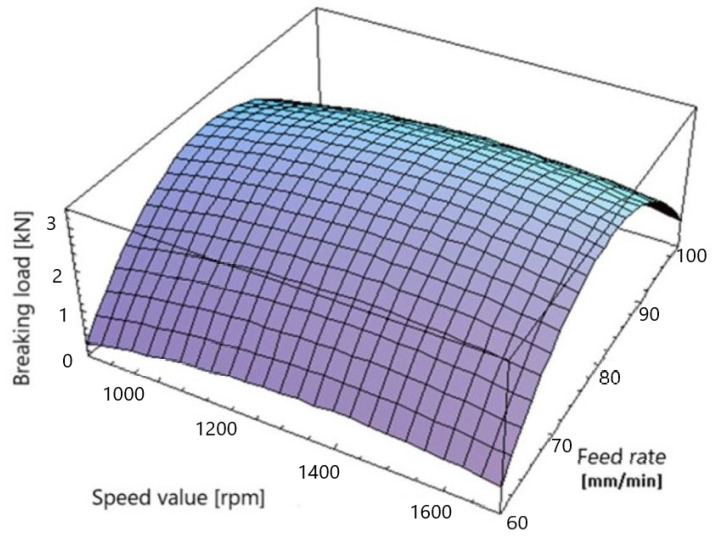
Decoding the regression equation.

**Table 1 materials-16-05116-t001:** Process parameters.

Rotation Speed (rpm)	Feed Rate (mm/min)
900	60
	80
	100
1300	60
	80
	100
1700	60
	80
	100

**Table 2 materials-16-05116-t002:** Mechanical properties of alloy 2024-T3 [33].

Condition	Tensile Strength (MPa)	Yield Stress (MPa)	Extension (%)	HBW (HB)
Pressing	490	380	8	130
Pulling	520	420	10	140

**Table 3 materials-16-05116-t003:** Chemical composition of alloy 2024 [33].

Alu/Stop 2024	Mg (%) 1.20–1.80	Mn (%) 0.30–0.90	Fe (%) ≤0.50	Si (%) ≤0.50	Si+Fe (%) -	Cu (%) 3.80–4.90	Zn (%) ≤0.25	Cr (%) ≤0.10	Mn+Cr (%) -
Ti (%) ≤0.15	Bi (%) -	Ni (%) -	Pb (%)	Sn (%) -	Zr (%) -	Zr+Ti (%) ≤0.20	Others (%) ≤0.05	Total others (%) ≤0.15	Al rest

**Table 4 materials-16-05116-t004:** Measurement results.

	Rotational Speed (RPM)	Feed Rate (mm/min)	F_1_ Breaking Force (kN)	F_2_ Breaking Force (kN)
Joining method, FSW	900	60	3.62	3.74
		80	3.80	3.90
		100	4.10	4.09
	1300	60	3.45	3.60
		80	3.40	3.19
		100	3.15	3.17
	1700	60	3.57	3.32
		80	3.70	3.78
		100	3.62	3.75

**Table 5 materials-16-05116-t005:** Test results.

Rotational Speed (RPM)	Feed Rate (mm/min)	F_1_ (kN)	F_2_ (kN)
900	60	3.62	3.74
	80	3.80	3.90
	100	4.10	4.09
1300	60	3.45	3.60
	80	3.40	3.19
	100	3.15	3.17
1700	60	3.57	3.32
	80	3.70	3.78
	100	3.62	3.75

**Table 6 materials-16-05116-t006:** A structured value matrix for the Hartley plan.

No.	Rotational Speed (RPM)	Feed Rate *p* (mm/min)
1.	1700	100
2.	1700	80
3.	1700	60
4.	1300	100
5.	1300	80
6.	1300	60
7.	900	100
8.	900	80
9.	900	60

**Table 7 materials-16-05116-t007:** Rupture force values and their averages.

No.	F_1_ (kN)	F_2_ (kN)	F Average (kN)
C1.	3.62	3.75	3.69
2.	3.70	3.78	3.74
3.	3.57	3.32	3.45
4.	3.15	3.17	3.16
5.	3.40	3.19	3.295
6.	3.45	3.60	3.5
7.	4.10	4.09	4.1
8.	3.80	3.90	3.85
9.	3.62	3.74	3.68

**Table 8 materials-16-05116-t008:** Hartley plan matrix.

No.	x_1_	x_2_	$x_{1}^{2}$	$x_{2}^{2}$	x_1_x_2_	ȳ
1.	+	+	+	+	+	3.69
2.	+	0	+	0	0	3.74
3.	+	−	+	+	−	3.45
4.	0	+	0	+	0	3.16
5.	0	0	0	0	0	3.295
6.	0	−	0	+	0	3.5
7.	−	+	+	+	+	4.1
8.	−	0	+	0	0	3.85
9.	−	−	+	+	+	3.68
Ʃ	−0.75	0.34	22.51	21.58	8.02	−

**Table 9 materials-16-05116-t009:** Measurement results, including measurement error.

No.	y_1_	y_2_	ȳ_i_	S^2^ (y_i_)	Ў_i_	(ȳ_i_ − Ў_i_)^2^
1.	3.62	3.74	3.69	0.0036	3.4578	0.0540
2.	3.80	3.90	3.74	0.0025	3.2278	0.2623
3.	4.10	4.09	3.45	0.000025	3.5695	0.0142
4.	3.45	3.60	3.16	0.005625	3.6789	0.2692
5.	3.40	3.19	3.295	0.011025	3.1267	0.1683
6.	3.15	3.17	3.5	0.0001	3.8976	0.1580
7.	3.57	3.32	4.1	0.015625	3.5689	0.2820
8.	3.70	3.78	3.85	0.0016	3.8798	0.0008
9.	3.62	3.75	3.68	0.004225	3.1674	0.2627

**Table 10 materials-16-05116-t010:** Welding parameters for best tensile strength values.

Rotational Speed (RPM)	Feed Rate (mm/min)	F_1_ (kN)	F_2_ (kN)
900	100	4.10	4.09

## Data Availability

Not applicable.

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
