# Peer review of "Advanced Analytical Methods of the Analysis of Friction Stir Welding Process (FSW) of Aluminum Sheets Used in the Automotive Industry"

_materials, 2023, doi:10.3390/ma16145116_

Round 1
Reviewer 1 Report
This is a timely effort by authors on"Energy consumption and CO2 emission of friction stir welding process of aluminum sheets used in the automotive industry". However, there are few suggestions to improve this manuscript as this manuscript has many jargons.
1. Novelty needs to be highlighted in a better way.
2. Introduction section contains mostly outdated references. No.of references must be latest I.e. published in last 5years as currently no. Of outdated references is more than 22.
3. Materials and Methods are shortly written and this section must be provided with all essential relevant details.
4. Results section needs much improvements in terms of following comments.
5. Fig. 2 must have dimensions showing the pin, shoulder, etc. sizes.
6. Fig. 3 is also ambiguous as it does not show the samples orientation. Please also show the sample geometry with dimensions.
7. I think there is no need of putting Table 3. You must mention the standard for tensile strength test and test speed must be replaced with crosshead speed.
8. What do you exactly mean by F1 and F2 in KN?
9. Figure no. of summary for best strength is wrongly mentioned and the bar graph representation must be replaced with any better form of scatter plot.
10. A picture of tensile tester may be inserted for better clarity.
11. As far as statistical tests are concerned, table 6 to table 9 must be appropriately explained for better clarity.
12. Each equation should also be explained in terms of the parameters in it with equation numbers.
13. The low CO2 emission justification based on the data collected by the National Balancing and Emissions 364Management Centre / Institute for Environmental Protection KOBiZE, in 2019 the emission factor, is not making sense. It is a very big claim that 310 grams of CO2 emission from FSW for 1m of weld length especially in the absence of any literature on it.
14. Plz write the future work after conclusion section.
Few grammar mistakes and sense deficiencies were observed which may be removed via better sentence structuring.
Author Response
I am sending answers to the Reviewer. Thank you very much for your comments, which allowed us to improve the article.
A. Generowicz

Reviewer 2 Report
The title of this manuscript is “Energy consumption and CO2 emission of friction stir welding 2 process of aluminum sheets used in the automotive industry”, however, the whole manuscript seems to have little relation with the title. The content and the title are two different things.
Rewrite the abstract part, “The main focus is on the strength of the friction stir welding connection and the energy consumption of the process to reduce CO2 emissions into the atmosphere.”. I donot see this is a main research content. Also, feed rate of 100 mm/min is the most optimal parameter, but the experiment only tried 60, 80 and 100 mm/min, the speed of 110 or 120 mm/min is better or worse than 100 mm/min?
Figure 1 is confusing.
There are more than one equation, please number them.
In line 360, “From the above analysis, the time required to join 1m of the test material is 10 minutes…” How long is the weld seam? If the simulation is correct, whether the efficiency is too low? Or the simulation has great variation?
Line 146, “comparison of the test results obtained with the average energy consumption for 146 conventional material joining methods”. Conventional joining method? Which method?
The whole introduction, the logic is messed up.
The title of this manuscript is “Energy consumption and CO2 emission of friction stir welding 2 process of aluminum sheets used in the automotive industry”, however, the whole manuscript seems to have little relation with the title. The content and the title are two different things.
Rewrite the abstract part, “The main focus is on the strength of the friction stir welding connection and the energy consumption of the process to reduce CO2 emissions into the atmosphere.”. I donot see this is a main research content. Also, feed rate of 100 mm/min is the most optimal parameter, but the experiment only tried 60, 80 and 100 mm/min, the speed of 110 or 120 mm/min is better or worse than 100 mm/min?
Figure 1 is confusing.
There are more than one equation, please number them.
In line 360, “From the above analysis, the time required to join 1m of the test material is 10 minutes…” How long is the weld seam? If the simulation is correct, whether the efficiency is too low? Or the simulation has great variation?
Line 146, “comparison of the test results obtained with the average energy consumption for 146 conventional material joining methods”. Conventional joining method? Which method?
The whole introduction, the logic is messed up.
Author Response

(The authors gave the same response as above.)

Round 2
Reviewer 1 Report
The authors have now improved the manuscript a lot. However, they have not provided the justification for energy consumption and CO2 emission in improved manuscript.
So they must work on it. After this minor correction, this manuscript can be accepted.
Author Response
We would like to thank you very much for giving us the opportunity to revise the paper and resubmit it for further scrutiny. We are thankful for all the suggestions, corrections and explanation asked by the reviewer for improving the quality of the manuscript. We have made as many changes as possible to implement the suggestions given by the reviewer. We have tried to clarify all the questions or doubts. The changes made in the manuscript are highlighted using green color. The Comments given by reviewer are presented below along with the response provided by the authors.
